# Development and Validation of the Patient-Centered Communication Competency Scale for Dental Hygienists

**DOI:** 10.3390/healthcare13111241

**Published:** 2025-05-24

**Authors:** Da-Eun Kim, Jong-Hwa Jang

**Affiliations:** 1Department of Public Health Science, Graduate School, Dankook University, Cheonan-si 31116, Republic of Korea; dek@dankook.ac.kr; 2Department of Dental Hygiene, College of Health Science, Dankook University, Cheonan-si 31116, Republic of Korea

**Keywords:** communication competency, dental hygienists, dental health service, oral health, patients

## Abstract

**Background/Objectives:** Communication skills are a core professional competency for dental hygienists. Accordingly, it is vital to develop a systematic scale that can objectively predict the relationship among communication skills and job satisfaction and various psychosocial factors. This study aimed to develop and validate the Patient-Centered Communication Competency Scale (PCCS) in dental hygienists for dental health service. **Methods:** Initial items were generated through a literature review, expert content validity assessment, and a preliminary survey. An online survey was conducted with 400 dental hygienists working in dental clinics and hospitals. Construct validity was examined using exploratory factor analysis (EFA) and confirmatory factor analysis (CFA), while reliability was assessed with Cronbach’s alpha. **Results:** The preliminary 38-item scale was refined through five rounds of EFA, resulting in an 11-item scale with three subscales: Assertiveness, Empathy, and Respect. CFA results indicated good model fit (χ^2^/df = 1.749, RMR = 0.027, RMSEA = 0.061, GFI = 0.941, NFI = 0.911, IFI = 0.960, CFI = 0.959, TLI = 0.949), demonstrating strong convergent and discriminant validity. The PCCS also demonstrated high internal consistency (Cronbach’s alpha = 0.862) and criterion validity, confirmed by its positive correlation with self-efficacy and job satisfaction. **Conclusions:** The proposed PCCS is a valid and reliable tool for assessing and improving dental hygienists’ communication skills, with potential applications in training programs and communication competency research in dental settings.

## 1. Introduction

Communication competency, defined as the ability to interact effectively with others using diverse methods, is a critical skill in modern society [1,2]. In healthcare, communication between professionals and patients is a fundamental aspect of medical practice. Recent changes in the healthcare landscape, along with increasing public awareness and expectations regarding healthcare services, have expanded the role of dental hygienists within dental institutions [3].

The role of dental hygienists varies depending on the national healthcare system, legal regulations, and professional recognition of their home country. Although communication is a core competency for dental hygienists, in Korea, it is only explanation-centered due to a lack of job autonomy and an institutional foundation [4]. On the other hand, the United States, Canada, and Northern Europe recognize the communication function of dental hygienists as a core professional competency through a patient-centered and prevention-centered approaches, actively reflecting this paradigm in their education and practice [4].

Dental care presents unique challenges, including financial burden, dental phobia, and the need for patient cooperation, making effective patient communication a critical factor [5]. For dental hygienists, strong communication skills are particularly important in areas such as counseling, patient management, and oral health education [6]. Among these skills, empathic communication plays a crucial role in patient-centered care, as it helps establish trust between healthcare providers and patients [7]. Studies have demonstrated that patient-centered communication enhances patient satisfaction and increases treatment adherence [1,8,9,10]. Furthermore, communication competency for dental hygienists influences both organizational effectiveness and job satisfaction [11,12,13].

Recognizing the significance of communication skills, the Korean Dental Hygienists Association and the Commission on Dental Accreditation of the American Dental Education Association emphasize them as a core competency for dental hygienists [14,15,16]. Communication is included in the Dental Middle Manager Job Competency Model, further underscoring its importance [17]. To meet these competency requirements, dental hygienists must adopt effective communication strategies, which first require a comprehensive understanding of their own communication skills. These skills must also be objectively measurable to ensure systematic and accurate evaluations.

Since communication interventions have broad effects, a consistent analytical framework is essential for evaluation [10]. Various communication competency scales have been developed and applied across different healthcare fields [18,19,20]. These include, first, the Global Interpersonal Communication Competence Scale (GICC) [18], which has an adequate number of items to properly reflect the necessary constructs and ensure content validity; second, the Nursing Assessment Communication Competence Scale (NACCS) for measuring the communication competency of clinical nurses [19]; and third, the Communication Ability Scale for assessing the communication skills of medical students and trained simulated patients [20].

In dentistry, a scale was developed to assess how patients perceive dental hygienists’ communication skills [10]; however, there remains a need for a self-assessment scale specifically designed to measure dental hygienists’ communication competency in their professional roles [21]. Higher patient-centered communication competency among dental hygienists is particularly important, as it can enhance self-efficacy, improve the quality of dental services, and ultimately increase job satisfaction. The development of the Patient-Centered Communication Competency Scale (PCCS) for dental hygienists is essential for evaluating their communication skills, developing training programs to enhance them, and conducting research on their relationship with various influencing factors. In this context, our study aimed to develop PCCS items based on empathic communication and assess their reliability and validity.

## 2. Materials and Methods

### 2.1. Design and Ethical Considerations

This study employed a methodological research design to test the reliability and validity of the PCCS in dental hygienists. To ensure ethical compliance, the study adhered to the principles of the Declaration of Helsinki, and its protocol was approved by the Institutional Review Board of Dankook University (DKU_2022-10-041 on 10 November 2022). All participants were informed of the study’s objectives and provided written informed consent before participation. The consent form outlined the study’s purpose, methodology, and key details concerning data anonymity, confidentiality, and the exclusive use of data for research purposes. Participants were also informed that they could withdraw from the study at any time without consequences. Only those who provided consent were included in the survey.

### 2.2. Scale Development

Our scale development process followed the framework proposed by DeVellis [22], consisting of two key stages: (1) conducting a literature review and expert content validation to generate PCCS items for dental hygienists, and (2) performing EFA and CFA to assess reliability and validity. This process involved the following steps: clarifying measurement concepts, generating scale items, determining the scale structure, conducting content validity assessments, reviewing and refining items, applying the scale, evaluating its effectiveness, and optimizing it for implementation (Figure 1).

#### 2.2.1. Conceptual Framework and Initial Item Generation

Communication competency is strongly emphasized as a core job skill for healthcare professionals [3], as provider–patient interactions—shaped by healthcare providers’ professionalism and their attitudes toward patients—play a crucial role in fostering patient trust, enhancing treatment effectiveness, and establishing and maintaining rapport [23].

In this study, patient-centered communication is defined as a dental hygienist’s ability to perceive and manage patient care through verbal and nonverbal interactions. Communication competency encompasses both grammatical and pragmatic knowledge, drawing upon language use in various social contexts [24]. Patient-centered communication encompasses five key elements: (1) empathic understanding, recognizing and reflecting patient’s emotions; (2) open dialog, involving active listening and meaningful feedback; (3) clear information provision; (4) shared decision-making with patients; and (5) gaining patient’s trust and approval of the proposed treatment.

According to empathic communication theory, empathy plays a crucial role in healthcare by enabling providers to understand and appropriately respond to patients’ emotions, particularly those with dental anxiety [7,8]. The present study builds on existing communication competency scales [1,5,11,13,19,25] and incorporates concepts from Rogers’ client-centered therapy to develop a comprehensive framework comprising items structured into three domains that represent the cognitive, affective, and behavioral components of communication [26,27,28]. The final items were designed to effectively assess communication competency among dental hygienists and were tested for validity and reliability (Figure 2).

The PCCS is expected to be a valuable tool for assessing and improving communication skills in both dental hygienists and dentists [11,20]. Unlike traditional information-based communication, patient-centered communication emphasizes respect for patients’ emotions, values, and needs, facilitating active patient participation in the decision-making process surrounding treatment. Respecting these concepts helps the patient and caregiver avoid misunderstandings that can lead to conflicts that are detrimental to both.

To identify and collate the initial scale items, we conducted a review of 196 domestic and international studies on communication competency published between 2013 and 2023. Based on this review, 90 preliminary items were selected, drawing on the Communication Ability Scale for Medical Students developed by Jeong and Kim [20] and the Communication Competency Scale for Nurses developed by Kim [5]. These items were then refined and adapted for self-assessment by dental hygienists working in dental clinics and hospitals. To enhance validity and reliability, two experts reviewed and refined the wording of each item, finalizing the initial scale.

#### 2.2.2. Content Validity

The content validity of the initial scale items was evaluated using a Delphi survey conducted with an expert panel comprising 10 professors and 8 dental hygienists. The professors are specialists in dental hygiene and dentistry with more than 10 years of work experience, while the dental hygienists hold a master’s degree or higher. All are currently employed in dental clinics and hospitals. Content validity was measured using the content validity index for items (I-CVI), with each item rated on a 4-point scale (1 = not relevant, 2 = somewhat relevant, 3 = quite relevant, 4 = highly relevant). The I-CVI for each item was determined by calculating the proportion of experts who assigned a rating of 3 or 4 [5].

A validity assessment was conducted using an I-CVI threshold of 0.80, as recommended by Ko and Hong [29]. Additionally, open-ended feedback from experts was incorporated to refine wording, eliminate redundant items, and remove items with low relevance. After three Delphi rounds, a final set of 38 items was selected, with an average I-CVI of 0.88.

#### 2.2.3. Preliminary Survey

We conducted a preliminary survey to assess readability, item comprehension, and questionnaire layout errors. Following Nunnally and Bernstein’s [30] recommendation that a sample size of 20–50 is appropriate for a preliminary survey, 20 dental hygienists actively practicing at dental clinics and hospitals were randomly selected.

The inclusion criteria were (1) dental hygienists working in clinics and hospitals who regularly communicate with patients and (2) those who understood the study’s purpose and voluntarily agreed to participate. The exclusion criteria were (1) healthcare professionals who were not dental hygienists, (2) dental hygienists who did not engage in patient communication, and (3) respondents who responded to less than 50% of the survey items.

Survey completion took approximately five minutes. All respondents indicated that the items were clear and easy to understand, necessitating no further modifications. The overall Cronbach’s α value was 0.922, confirming strong internal consistency.

### 2.3. Validity and Reliability

#### 2.3.1. Participants

For participant selection, we applied the same inclusion and exclusion criteria as in the preliminary survey. The sample size was determined based on established guidelines, which recommend a minimum of 100 participants for EFA [31] and 10 to 20 times the number of factors for CFA [32]. Accordingly, a total of 400 participants were randomly selected, with 200 allocated to EFA and 200 to CFA.

#### 2.3.2. Measurement

The PCCS for dental hygienists consisted of 38 items, finalized through a content validity assessment from an expert panel and a preliminary survey (Appendix A). Sociodemographic characteristics included sex, age, education, workplace, work experience, and current position.

Criterion validity was assessed using the 24-item Dental Hygienist Job Satisfaction Scale developed by Park [33]. Each item on this measure is rated on a 5-point Likert scale (1 = not at all, 5 = very much), with higher scores indicating greater job satisfaction. Cronbach’s α for this instrument was 0.910 in Park’s [33] study and 0.933 in this study.

General self-efficacy was measured using an eight-item scale originally developed by Chen et al. [34] and adapted by Cho [35]. Items were rated on a 5-point Likert scale (1 = not at all, 5 = very much), with higher scores indicating greater self-efficacy. Cronbach’s α for this scale was 0.844 in Cho’s study [35] and 0.904 in this study.

#### 2.3.3. Data Collection

Data were collected through an online survey conducted between 27 March and 16 April 2023. We randomly selected 50 dental clinics and hospitals in South Korea for inclusion in the study. Through the collaboration of the community dental hygiene society, the survey included an information sheet detailing the study’s purpose, procedure, and estimated completion time. Of the 417 responses received, 17 were excluded owing to incomplete or inconsistent answers, resulting in a final dataset of 400 responses (95.9%) for analysis.

#### 2.3.4. Validity and Reliability Tests

Data analysis was performed using IBM SPSS 23.0 and AMOS 23.0. Descriptive statistics were presented as mean, standard deviation, frequency, and percentage. Normality was assessed based on skewness (<2) and kurtosis (<7) [36]. Normal distribution was confirmed for all items, with skewness ranging from −0.263 to −0.981 and kurtosis from −0.387 to 1.51. Additionally, internal consistency was evaluated by examining item-total correlation coefficients, with values below 0.30 indicating low internal consistency within the scale [37]; all correlation coefficients were high.

Construct validity was assessed through EFA and CFA. EFA was conducted using Bartlett’s test of sphericity and the Kaiser–Meyer–Olkin (KMO) measure, followed by principal component analysis with varimax rotation. Factors were retained based on eigenvalues greater than 1 and a cumulative variance explained of at least 50–60% [31]. A factor loading threshold of 0.4 was applied [38].

CFA evaluated model fit using absolute and incremental fit indices. Convergent validity was assessed by calculating the average variance extracted (AVE) and construct reliability (CR) [39]. Model fit indices were interpreted based on the following criteria: for CFA model fit, a root mean residual (RMR) ≤0.05 was considered good and a root mean square error of approximation (RMSEA) ≤0.05 was classified as excellent, ≤0.08 as acceptable, and ≤0.10 as moderate [40]. Additionally, values ≥0.90 were considered very good and values ≥0.80 were deemed acceptable for the goodness-of-fit index (GFI), comparative fit index (CFI), normed fit index (NFI), Tucker–Lewis index (TLI), and incremental fit index (IFI) [41]. Convergent validity was considered established when the following criteria were met: standardized factor loadings ≥ 0.50, critical ratio ≥ 1.96 (*p* < 0.05), AVE ≥ 0.50, and CR ≥ 0.70 [42]. Discriminant validity, which indicates that measures of different constructs exhibit low correlations, was assessed using correlation coefficients and the square root of the AVE (√AVE) [43].

Job satisfaction and self-efficacy were used for criterion validity. According to the Job Characteristics Theory, job satisfaction is enhanced by meaningful interpersonal relationships experienced in executing the job [44]. According to Social Cognitive Theory, self-efficacy is formed by an individual’s successful experiences, feedback, and psychological state, for which communication skills can act as a major catalyst [45]. Therefore, job satisfaction and self-efficacy are external variables closely related to the concept of patient-centered communication skills, making them appropriate criteria for evaluating the criterion validity of this scale. Criterion validity was examined by analyzing the correlations between the PCCS and measures of job satisfaction and self-efficacy. A correlation coefficient range of 0.40–0.80 was considered appropriate for criterion-related validity [22]. Internal consistency was assessed using Cronbach’s α.

### 2.4. Optimization of PCCS

To optimize the PCCS with established validity and reliability, we consulted a professor of Korean language education to review the grammar, readability, and overall clarity of the final items. The review confirmed that the items were free of grammatical errors and were suitable for assessing the communication competency of dental hygienists, requiring no further modifications.

## 3. Results

### 3.1. Demographic Characteristics of Study Participants

Participants in their 30s formed the largest age group (n = 212, 53%), and a majority of the total sample were female (n = 362, 90.5%). Most participants held associate degrees (n = 222, 55.5%), followed by bachelor’s degrees (n = 132, 33%) and master’s degrees or higher (n = 46, 11.5%). The most common workplace setting was private dental clinics (n = 300, 75.0%), and most participants worked as clinical dental hygienists (n = 257, 64.3%) (Table 1).

### 3.2. Item Analysis

Item analysis was conducted using the mean and standard deviation of each item. With skewness ranging from −0.263 to −0.981 and kurtosis from −0.387 to 1.51, normality was confirmed for all items. The corrected item-total correlation coefficient for the scale ranged from 0.948 to 0.951. The overall Cronbach’s α value for the scale was 0.951, indicating strong internal consistency.

### 3.3. Construct Validity

#### 3.3.1. Exploratory Factor Analysis (EFA)

EFA was performed five times using eigenvalues as a criterion, with a factor loading threshold set at 0.4. The five EFAs were as follows:

In the first EFA, seven items exhibiting double loadings on multiple factors and two items with factor loadings below 0.4 were removed, resulting in 29 items being retained. In the second EFA, five items exhibiting double loadings on multiple factors and one item with a communality value below 0.4 were removed, resulting in 23 items being retained. In the third EFA, five items exhibiting double loadings on multiple factors and one item with a factor loading below 0.4 were removed, resulting in 17 items being retained. In the fourth EFA, three items exhibiting double loadings on multiple factors and three items with a communality value below 0.4 were removed, resulting in 11 items being retained.

In the fifth EFA, the KMO coefficient was 0.84, and Bartlett’s test of sphericity was significant (χ^2^ = 674.772, *p* < 0.001), with a cumulative variance explained of 60.51%. The final 11 items had communality values above 0.4, maximum factor loadings above 0.7, and no items with double loadings. The items were categorized into three factors (Table 2).

The three final factors were reviewed and labeled as follows: Factor 1—“Respect”, referring to listening, providing feedback, and building trust with patients; Factor 2—“Assertiveness”, representing the process of clarifying communication for effective patient interactions; and Factor 3—“Empathy”, reflecting the ability to understand patient expectations and perspectives, engage in discussion, and negotiate for relationship-oriented communication in healthcare.

#### 3.3.2. Confirmatory Factor Analysis for Validity (CFA)

In model fit assessment, while a significant chi-square test (χ^2^ = 71.721, *p* < 0.05) may suggest a discrepancy between the model and observed data, the χ^2^/df ratio was 1.749, which falls within the acceptable range, indicating an adequate model fit. Other model fit indices also satisfied their respective thresholds: RMSEA = 0.061, TLI = 0.949, RMR = 0.027, GFI = 0.941, NFI = 0.911, RFI = 0.880, IFI = 0.960, and CFI = 0.959 (Table 3).

For convergent validity, all standardized lambda (λ) values exceeded 0.50. AVE values were 0.533 for Assertiveness, 0.472 for Empathy, and 0.614 for Respect. CR values were 0.818 for Assertiveness, 0.728 for Empathy, and 0.864 for Respect, all meeting the convergent validity criteria (Table 3 and Figure 3).

For discriminant validity, all factor correlation coefficients ranged from 0.551 to 0.584, which were lower than the corresponding √AVE values (0.687–0.783), confirming that the discriminant validity criteria were met (Table 4).

### 3.4. Criteria Validity and Reliability of PCCS

The final PCCS items retained in this study are presented in Appendix A. The PCCS level by demographic characteristics was higher for male than for female participants (*p* = 0.043), while those with a master’s degree or higher had a higher level than those with a bachelor’s degree or lower (*p* = 0.001). In addition, PCCS was higher in those aged 40 years or older (*p* = 0.001). There was no difference in PCCS by length of service, position, or work location, as shown in Appendix A.

To assess criterion validity, correlations between the PCCS and measures of self-efficacy and job satisfaction were examined using data from the full analysis set (n = 400; Table 5). The PCCS demonstrated strong positive correlations with both self-efficacy (r = 0.603) and job satisfaction (r = 0.624). Additionally, correlations between the PCCS subscales and their corresponding criterion measures were also positive, further supporting criterion validity.

The internal consistency of the PCCS, as measured by Cronbach’s α, was 0.862 for the overall scale, and, when broken down into subscales, was 0.731 for Assertiveness, 0.701 for Empathy, and 0.771 for Respect, indicating satisfactory reliability (Table 5).

## 4. Discussion

This study aimed to develop and validate the PCCS for dental hygienists. To achieve this, 38 assessment items were extracted from the existing literature and categorized into cognitive, affective, and behavioral domains [26]. The scale components included factors such as effective communication and presentation skills, active listening and comprehension, and discussion and negotiation skills—each reflecting self-expression, acceptance of others’ opinions, and interaction and coordination with others [46]. Through five rounds of EFA, the final model was refined to include 11 items structured into three factors (subscales): Respect, Assertiveness, and Empathy. These 11 items encapsulate the essential communication competencies that dental hygienists should possess.

The PCCS represents a communication approach that respects patients’ values, preferences, and needs, encouraging their active participation in health and treatment-related decision-making. Rather than merely transmitting information, the PCCS emphasizes understanding patients’ emotions and expectations while fostering collaborative decision-making throughout the treatment process. The patient–dental hygienist relationship is grounded in human dignity, and within this interaction, empathic communication, based on person-centered care theory, plays a crucial role [7,26].

Factor 1 of the PCCS is Respect. This refers to the ability to focus on a patient’s words, utilize verbal and nonverbal cues for understanding, and express acknowledgment and empathy toward their emotions and concerns. The four items in this subscale assess whether these skills are effectively conveyed. Notably, when healthcare providers use overly complex or lengthy explanations, patients may perceive this as a lack of respect. Therefore, providing information at an appropriate level and verifying the patient’s understanding fosters trust and conveys respect [39,47]. To further strengthen the communication skills of dental hygienists, it is necessary to operate programs that can increase knowledge of dental information [48].

The inclusion of the item “Encourage patients to express their emotions freely” under this subscale is supported by previous research, which suggests that when dental hygienists establish trust as oral health professionals, they foster positive therapeutic relationships and enhance communication effectiveness [1].

Factor 2, Assertiveness, reflects dental hygienists’ ability to explain treatment processes and options clearly and comprehensibly, empowering patients to make informed decisions. The four items in this subscale assess dental hygienists’ ability to encourage patients’ proactive participation in developing treatment plans that align with their values and preferences. The item “Summarize key points throughout the conversation” aligns with previous research, indicating that providing essential information, including potential risks and symptoms, can mitigate medical disputes and strengthen patient trust in healthcare providers [10].

Factor 3, Empathy, represents dental hygienists’ ability to recognize patients’ anxiety and fear, provide psychological reassurance, and maintain a supportive attitude throughout the treatment process. The three items in this subscale assess communication strategies that help alleviate patient concerns and manage expectations. The inclusion of the item “Ask if the patient has any additional questions or concerns” aligns with prior research, which highlights that acknowledging patient discomfort strengthens trust-based relationships between dental hygienists and their patients [49].

The PCCS developed in this study was validated through CFA, confirming model fit, convergent validity, and discriminant validity. Criterion validity was established through strong positive correlations between the PCCS and measures of job satisfaction and self-efficacy, and internal consistency was confirmed with Cronbach’s α of 0.862. These findings align with previous studies that highlight the positive correlation among communication competence, self-efficacy, and job satisfaction [50,51,52,53]. Specifically, nurses with high communication self-efficacy showed better performance and job satisfaction in various work tasks [50]. Moreover, communication skills and nursing work environment were identified as important factors influencing nurses’ job satisfaction [51], confirming the importance of clinical nurses’ communication skills for efficient human resource management [52]. In addition, key components of effective communication skills training in dental education include motivational interviewing, open-ended questioning, affirmations, reflective listening, and summaries to enhance patient engagement and adherence to treatment plans [53].

Despite the inherent overlap and conceptual ambiguity among communication components, the PCCS successfully achieved discriminant validity, which is a notable strength of this study. Therefore, the PCCS is expected to serve as a valuable tool for assessing the communication competency of dental hygienists.

### Strengths and Limitations

Time constraints, difficulties in establishing rapport, the oral health illiteracy of the patients, in addition to poor communication skills, perceptions, and language barriers, often hinder communication between dental hygienists and patients [53]. The PCCS developed in this study was specifically designed for dental hygienists working in dental clinics and hospitals. Unlike existing tools, the PCCS evaluates the interactional outcomes of communication between dental hygienists and patients rather than focusing solely on individual communication skills [10]. Additionally, the PCCS was designed as a concise 11-item scale that incorporates essential communication competencies for dental hygienists, making it both easy to understand and practical for clinical use. The PCCS serves as a foundational tool for research on improving communication skills in dental hygienists and can facilitate the development of training programs. It can also function as an educational and assessment tool for further research aimed at developing training programs to enhance patient satisfaction and treatment adherence. Furthermore, the scale has potential applications in structural equation modeling and theory validation studies related to communication skills in dental hygienists.

This study has some limitations. Its generalizability is limited, as participants were restricted to dental hygienists in clinics and hospitals. Communication strategies may vary when interacting with special populations, such as patients with disabilities, long-term care facility residents, or home-based oral care recipients [39]. In addition, it is necessary to confirm the usefulness of the PCCS according to the job characteristics of dental hygienists in different workplaces.

Future research should explore the development of additional communication competency scales that account for the diverse roles of dental hygienists and the specific needs of different patient populations. Moreover, intervention and longitudinal studies investigating the impact of patient-centered communication on dental fear and anxiety would provide valuable insights for shaping policies that enhance patient-centered care in dental clinics and hospitals.

In summary, the PCCS is expected to contribute to the advancement of patient-centered care in dental practice and, ultimately, to improving patients’ oral health.

## 5. Conclusions

The PCCS developed in this study consists of 11 items categorized into the Respect, Assertiveness, and Empathy subscales. Its reliability and content, construct, and criterion validity were successfully established. This tool is distinct from existing measures, as it emphasizes empathic interactions and relationship-building between dental hygienists and patients in clinical settings. Additionally, the PCCS can be used as a self-assessment tool to help dental hygienists enhance their communication skills. By improving communication competency, dental hygienists may experience greater job satisfaction and contribute to better patient care. As a concise and efficient tool, the PCCS is easy to administer and can be widely applied in research studies on communication in dental hygiene practice. Furthermore, its applicability extends to training programs and performance assessments, supporting the development of patient-centered care models in dentistry.

## Figures and Tables

**Figure 1 healthcare-13-01241-f001:**
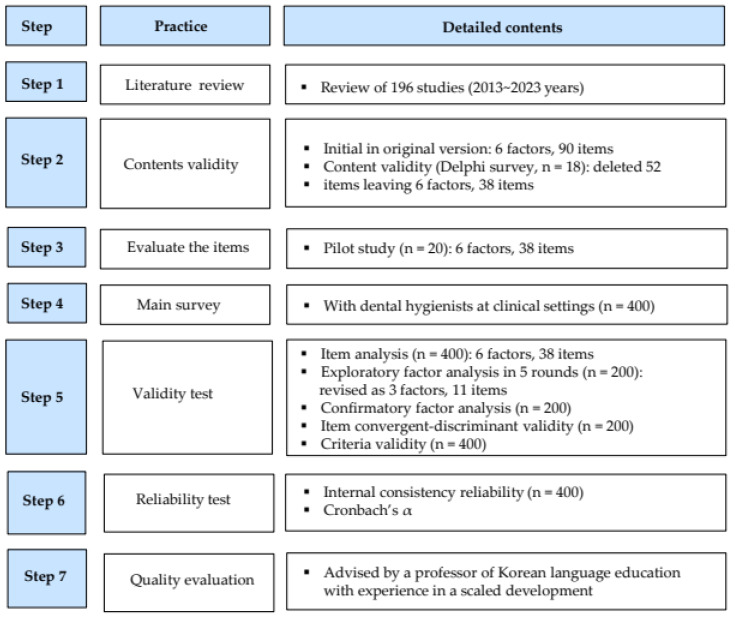
Process of development for Patient-Centered Communication Competency Scale (PCCS).

**Figure 2 healthcare-13-01241-f002:**
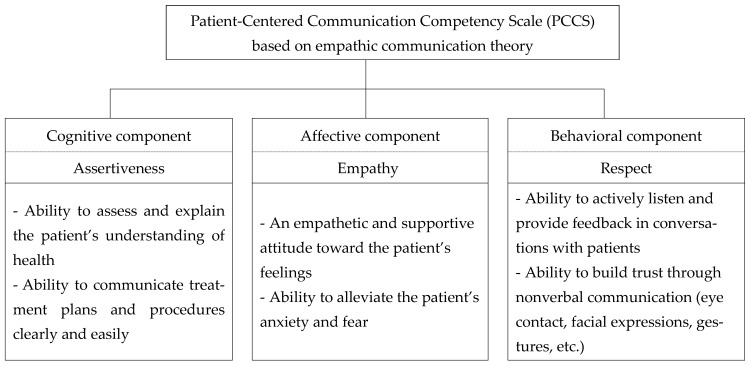
Conceptual framework.

**Figure 3 healthcare-13-01241-f003:**
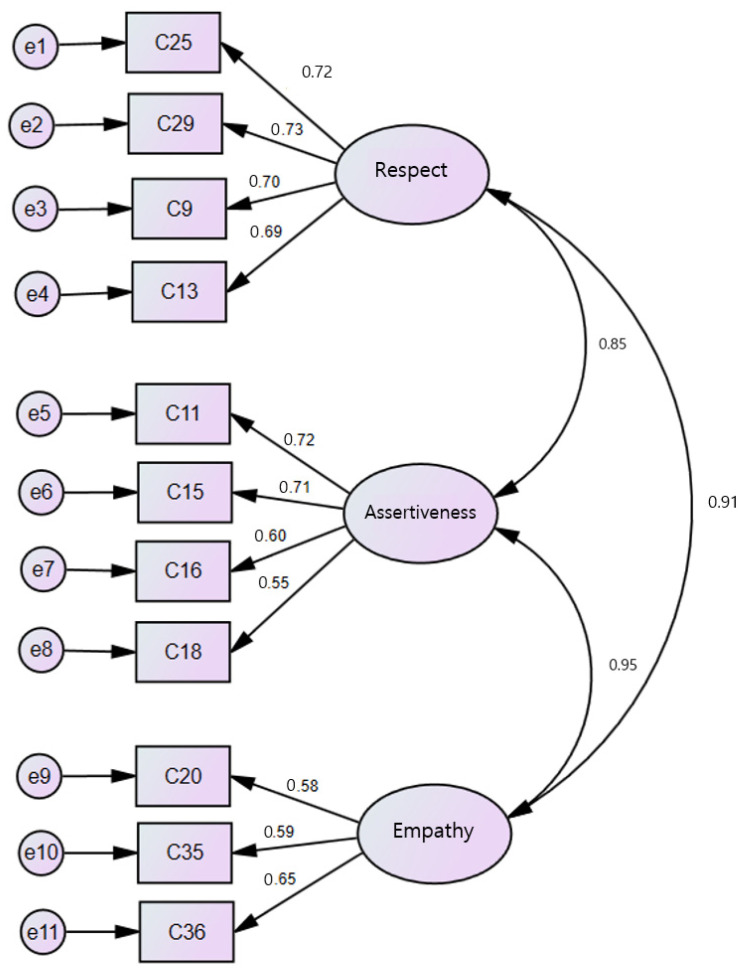
Path diagram of final model for Patient-Centered Communication Competency Scale (PCCS). AVE ≥ 0.472; CR ≥ 0.728.

**Table 1 healthcare-13-01241-t001:** Demographic characteristics of study participants.

Variables	Categories	Total (n = 400)	EFA (n = 200)	CFA (n = 200)
		n (%)	n (%)	n (%)
Sex	Male	38 (9.5)	18 (9.0)	20 (10.0)
	Female	362 (90.5)	182 (91.0)	180 (90.0)
Age (in years)	≤29	147 (36.8)	75 (37.5)	72 (36.0)
	30–39	212 (53.0)	103 (51.5)	109 (54.5)
	≥40	41 (10.3)	22 (11.0)	19 (9.5)
Education	College	222 (55.5)	112 (56.0)	110 (55.0)
	University	132 (33.0)	67 (33.5)	65 (32.5)
	≥Master	46 (11.5)	21 (10.5)	25 (12.5)
Workplace	Dental clinic	300 (75.0)	143 (71.5)	157 (78.5)
	Dental hospital	84 (21.0)	48 (24.0)	36 (18.0)
	Tertiary hospital	13 (3.3)	8 (4.0)	5 (2.5)
	Others	3 (0.8)	1 (0.5)	2 (1.0)
Work experience (in years)	≤3	70 (17.5)	32 (16.0)	38 (9.0)
	≥3–≤7	140 (35.0)	69 (34.5)	71 (35.5)
	≥7	178 (44.5)	94 (47.0)	84 (42.0)
Current position	Clinical dental hygienist	257 (64.3)	129 (64.5)	128 (64.0)
	Team leader	53 (13.3)	29 (14.5)	24 (12.0)
	Counseling manager	68 (17.0)	31 (15.5)	37 (18.5)
	Others	22 (5.5)	11 (5.5)	11 (5.5)

**Table 2 healthcare-13-01241-t002:** Final exploratory factor analysis results (n = 200).

No	Items	Item-Total Correlation	Factor Loading		
			1	2	3
25	Encourage patients to express their emotions freely	0.612	0.743	0.093	0.226
29	Present information by topic at a pace the patient can follow	0.600	0.763	0.107	0.080
9	Mirror the patient’s words or actions to demonstrate empathy	0.587	0.692	0.272	0.185
13	Repeat questions to clarify implied content and emotions	0.502	0.661	0.198	0.157
11	Communicate in a structured and proficient manner	0.462	0.361	0.526	0.235
15	Summarize key points throughout the conversation	0.683	0.161	0.791	0.176
16	Clearly signal transitions between topics or situations.	0.628	0.103	0.695	0.367
18	Redirect off-topic discussions back to the main topic	0.553	0.164	0.725	−0.011
20	Clarify expectations regarding diagnosis, treatment, prognosis	0.682	0.165	0.302	0.751
35	Conclude communication with gratitude for cooperation	0.675	0.151	0.104	0.801
36	Ask if the patient has any additional questions or concerns	0.672	0.256	0.118	0.770
Eigenvalues			2.360	2.160	21.453
% of variance			21.453	19.640	41.093
% of cumulated variance			21.453	19.413	60.508
Kaiser–Meyer–Olkin (KMO) value: 0.84, Bartlett’s sphericity test value: 674.772 (*p* < 0.001)					

**Table 3 healthcare-13-01241-t003:** Confirmatory factor analysis results (n = 200).

Concept	Factors	Item Number	Standardized Estimate (β)	SE	AVE	CR
Behavioral	Respect	25	0.715	0.308	0.614	0.864
		29	0.729	0.273		
		9	0.703	0.339		
		13	0.691	0.348		
Cognitive	Assertiveness	11	0.715	0.342	0.533	0.818
		15	0.705	0.327		
		16	0.598	0.318		
		18	0.547	0.473		
Affective	Empathy	20	0.583	0.441	0.472	0.728
		35	0.594	0.440		
		36	0.646	0.362		
Model fitness: χ^2^/df = 1.749, RMR = 0.027, RMSEA = 0.061, GFI = 0.941, NFI = 0.911, IFI = 0.960, CFI = 0.959, TLI = 0.949						

SE = standard error; AVE = average variance extracted; CR = construct reliability; RMR = root mean square residual; RMSEA = root mean square error of approximation; GFI = goodness-of-fit index; NFI = normed fit index; IFI = incremental fit index; CFI = comparative fit index; TLI = Tucker–Lewis index.

**Table 4 healthcare-13-01241-t004:** Correlation and discriminant validity (n = 200).

Factors	Respect	Assertiveness	Empathy
Respect	0.783		
Assertiveness	0.584 *	0.730	
Empathy	0.551 *	0.566 *	0.687

* *p* < 0.001; shaded section = discriminant validity; non-shaded section = correlation.

**Table 5 healthcare-13-01241-t005:** Correlation between PCCS and job satisfaction for criterion-related validity (n = 400).

PCCS	Self-Efficacy	Job Satisfaction	Mean ± SD	Cronbach’s α
	r	r		
Total	0.603 *	0.624 *	3.94 ± 0.52	0.862
Respect	0.441 *	0.592 *	3.94 ± 0.63	0.777
Assertiveness	0.526 *	0.520 *	3.95 ± 0.57	0.731
Empathy	0.548 *	0.452 *	3.93 ± 0.66	0.701

* *p* < 0.01; SD = standard deviation; PCCS = Patient-Centered Communication Competency Scale.

## Data Availability

The data presented in this study are available on reasonable request from the corresponding author.

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
