# Peer review of "Development and Validation of the Patient-Centered Communication Competency Scale for Dental Hygienists"

_healthcare, 2025, doi:10.3390/healthcare13111241_

Round 1
Reviewer 1 Report
Comments and Suggestions for Authors
I thank the editor for considering me to review this paper.
The study has been done well and it is in a relevant area. However, there are some suggestions that the authors may consider :
- Given the nature of communication competencies, an oblique rotation such as Promax may be more theoretically appropriate. Authors should justify their choice or revise accordingly.
- Only relying on the eigenvalues greater than 1 can inflate factor retention. Authors should also take into consideration the scree plot analysis or parallel analysis to support the number of factors extracted.
- Authors mention 0.4 factor loading threshold. However, it is unclear whether any items were removed during EFA. Authors should specify if item deletion occurred and report the number of items retained.
- Authors should elaborate on the theoretical basis for selecting job satisfaction and self-efficacy as external constructs for assessing criterion validity in their study.
- Since only one language reviewer is mentioned, involving multiple experts is suggested for content validation. This may be mentioned as a limitation.
Author Response
Our point-by-point response to the reviewer’s comments and suggestions is listed below:
We thank you for taking the time and effort necessary to review our manuscript and provide us with these valuable comments and suggestions. Accordingly, we revised our manuscript and made changes to it. Please note that changes to the manuscript are highlighted in yellow for your convenience.
I thank the editor for considering me to review this paper.
The study has been done well and it is in a relevant area. However, there are some suggestions that the authors may consider:
(point 1) Given the nature of communication competencies, an oblique rotation such as Promax may be more theoretically appropriate. Authors should justify their choice or revise accordingly.
(Response to point 1) Thank you for your insightful comments. The PCCS developed in this study is based on the empathy theory and consists of items that include cognitive (Assertiveness), affective (Empathy), and behavioral (Respect) domains, and they are clearly separated and operate independently. The main purpose of the PCCS is to develop a measurement tool for dental hygienist education, evaluation, and practical application. The PCCS is a measurement tool composed of clear and separated domains rather than a complex factor structure. Please understand that Varimax rotation was applied in our study because it rotates the items so that they load clearly on only one factor, making the interpretation of the results simple and clear.
(point 2) Only relying on the eigenvalues greater than 1 can inflate factor retention. Authors should also take into consideration the scree plot analysis or parallel analysis to support the number of factors extracted. Authors mention 0.4 factor loading threshold. However, it is unclear whether any items were removed during EFA. Authors should specify if item deletion occurred and report the number of items retained.
(Response to point 2) We agree with your constructive comments. We reviewed the results of the scree plot analysis to review the number of extracted factors. In addition, the results of the five EFA analyses were presented in detail in the research results section. The initial 36 items were finally derived as 29 items in Round 1, 23 items in Round 2, 17 items in Round 3, and 11 items in Rounds 4 and 5 by deleting items with factor loadings less than 0.4. The overall tool development process was added as a Figure 1.
Figure 1. Process of development for Patient-Centered Communication Competency Scale (PCCS)
(point 3) Authors should elaborate on the theoretical basis for selecting job satisfaction and self-efficacy as external constructs for assessing criterion validity in their study.
(Response to point 3) We agree with your meaningful comment and have added the following:
Job satisfaction and self-efficacy were used for criterion validity. According to the Job Characteristics Theory, meaningful interpersonal relationships experienced during the job performance process act as factors that enhance job satisfaction [44]. Self-efficacy is formed by an individual's successful experiences, feedback, and psychological state in the Social Cognitive Theory, and communication skills can act as a major catalyst for these positive experiences [45]. Therefore, job satisfaction and self-efficacy are external variables closely related to the concept of patient-centered communication skills, and can be appropriate criteria for evaluating the criterion validity of this scale.
Oldham, G. R.; Hackman, J. R. Not what it was and not what it will be: The future of job design r
- Bandura, Albert "Self-Efficacy", The Corsini Encyclopedia of Psychology, American Cancer Society, 2010, pp. 1–3.
(point 4) Since only one language reviewer is mentioned, involving multiple experts is suggested for content validation. This may be mentioned as a limitation.
(Response to point 4) Thank you for your detailed comments. In the PCCS development process, after the initial items were determined, a Delphi survey was conducted by forming a panel of 18 experts to evaluate the content validity of each item. (Refer to “2.2.2. Content Validity”)
In addition, in order to optimize the 11 items that were finally derived, a final review was conducted by a professor of Korean language education to check the grammar and readability of the items. Reflecting your opinion, “Limitations and Future Research Directions” added in the Discussion section as follows:
In addition, it is necessary to confirm the usefulness of the PCCS according to the job characteristics of dental hygienists at each workplace.
Reviewer 2 Report
Comments and Suggestions for Authors
Review.
Journal Healthcare (ISSN 2227-9032)
Manuscript ID healthcare-3583126
Type Article
Title Development and Validation of the Patient-Centered Communication Competency Scale for Dental Hygienists
Authors Da-Eun Kim , Jong-Hwa Jang *
Section Nursing
Special Issue Oral Health Care and Services for Patients
This article presents a study aimed at developing a competency scale for dental hygienists to improve their communication skills with patients. This work seeks to develop items based on empathic communication and to evaluate their reliability and validity.
General considerations
- In its introduction, the article should emphasize the global disparity in the role of hygienists in the management of oral health care. The role of hygienists is nonexistent in some countries, while it is predominant in others.
2- The use of computers should be more mentioned in the article. Indeed, computers, in the context of this study in a hospital and clinical setting, present advantages. Access to results on the general condition of patients, thanks to computers, is facilitated for dental hygienists working in clinical and hospital establishments. This allows for improvements directly with patients. Dental informatique (Felix Gomez GG, Mao JM, Thyvalikakath TP, Li S. Building Bridges: Fostering Collaborative Education in Training Dental Informaticians. Appl Clin Inform. 2025 Jan).
- The use of a recent bibliography is desirable to update the article and attract readers.
Important details considerations
Line 20 : Abbreviations appearing for the first time should be translated in full for better understanding. ( RMR, RMSEA, GFI, NFI, IFI, CFI, TLI)
Line 48 : add more recent articles like : (Sasaki N, Pang J, Surdu S, Morrissey RW, Vujicic M, Moore J. Workplace factors associated with job satisfaction among dental hygienists and assistants in the United States. Health Aff Sch. 2025 ) and
(Loeffen AWM, Van Swaaij BWM, Saminsky M, Slot DE. Common practices of dental implant maintenance among dental hygienists working in the Netherlands - A survey. Int J Dent Hyg. 2025 Feb). Line 104 : the fifth point, (5) “trust building, which aims to establish rapport to improve treatment satisfaction”. May be modified as follows… “gain the patient's trust and approval of the proposed treatment (avoiding understating or overstating the expected results). We are not looking for "treatment satisfaction" but patient satisfaction. This is different and requires adaptation at several levels (socio-economic, mental, physiological, etc.). Line 119: A note can be added as follows: "Respecting these concepts helps to avoid misunderstandings between the patient and the caregiver which can lead to conflicts that are detrimental to both."
Line 315 : Reference (23) is too old .
Line 235 : Considering the particularities of the study participants, it seems interesting to specify whether they are general practitioners or specialists in periodontology, implantology, surgery, orthodontics, etc.? Lines 301-303 : Who are we talking about, to validate the evaluation of the criterion, the correlation between the PCCS and the measurements carried out? Who is concerned with self-efficacy? Who is concerned with job satisfaction? Should the degree of long-term patient satisfaction rather be the indicator of the success of this preliminary investigation?
Author Response
Our point-by-point response to the reviewer’s comments and suggestions is listed below:
We thank you for taking the time and effort necessary to review our manuscript and provide us with these valuable comments and suggestions. Accordingly, we revised our manuscript and made changes to it. Please note that changes to the manuscript are highlighted in yellow for your convenience.
This article presents a study aimed at developing a competency scale for dental hygienists to improve their communication skills with patients. This work seeks to develop items based on empathic communication and to evaluate their reliability and validity.
General considerations
(point 1) In its introduction, the article should emphasize the global disparity in the role of hygienists in the management of oral health care. The role of hygienists is nonexistent in some countries, while it is predominant in others.
(Response to point 1) We agree with the constructive comments. We have added the current status of the role of dental hygienists to the introduction section as follows:
The role of dental hygienists varies depending on the national health care system, legal regulations, and professional recognition. Communication is a core competency of dental hygienists, but in Korea, it is limited to explanation-centered due to lack of job autonomy and institutional foundation [4]. On the other hand, the United States, Canada, and Northern Europe recognize the communication function of dental hygienists as a core professional role through a patient-centered and prevention-centered approach and actively reflect it in education and practice [4].
- Kim, A.; Cho, M. A comparison of dental hygienists’ legal duties among nations: Korea, USA and Canada. J. Korean Soc. Oral Health Sci. 2019, 7(1), 18-28. https://doi.org/10.33615/jkohs.2019.7.1.18
(point 2) The use of computers should be more mentioned in the article. Indeed, computers, in the context of this study in a hospital and clinical setting, present advantages. Access to results on the general condition of patients, thanks to computers, is facilitated for dental hygienists working in clinical and hospital establishments. This allows for improvements directly with patients. Dental informatique (Felix Gomez GG, Mao JM, Thyvalikakath TP, Li S. Building Bridges: Fostering Collaborative Education in Training Dental Informaticians. Appl Clin Inform. 2025 Jan).
The use of a recent bibliography is desirable to update the article and attract readers.
(Response to point 2) We agree with the meaningful comments and added the need for dental informatics based on prior research in the discussion section as follows:
To further strengthen the communication skills of dental hygienists, it is necessary to operate programs that can increase knowledge of dental information [49].
- Felix Gomez, G.G.; Mao, J.M.; Thyvalikakath, T.P.; Li, S. Building bridges: Fostering collaborative education in training dental informaticians. Appl. Clin. Inform. 2025, 16(1), 205-214. https://doi.org/10.1055/a-2446-0515
Important details considerations
(point 3) Line 20 : Abbreviations appearing for the first time should be translated in full for better understanding. ( RMR, RMSEA, GFI, NFI, IFI, CFI, TLI)
(Response to point 3) Thank you for your thoughtful comments. RMR, RMSEA, GFI, NFI, IFI, CFI, and TLI, which are the model fit tests in CFA, are commonly used terms, so only abbreviations are used in the abstract for brevity. However, please understand that the full names are provided in the text.
(point 4) Line 48 : add more recent articles like : (Sasaki N, Pang J, Surdu S, Morrissey RW, Vujicic M, Moore J. Workplace factors associated with job satisfaction among dental hygienists and assistants in the United States. Health Aff Sch. 2025 ) and (Loeffen AWM, Van Swaaij BWM, Saminsky M, Slot DE. Common practices of dental implant maintenance among dental hygienists working in the Netherlands - A survey. Int J Dent Hyg. 2025 Feb).
(Response to point 4) We have added the latest literature you suggested below.
- Sasaki, N.; Pang, J.; Surdu, S.; Morrissey, R.W.; Vujicic, M.; Moore, J. Workplace factors associated with job satisfaction among dental hygienists and assistants in the United States. Health Aff. Sch. 2025, 3(1), qxae 147. https://doi.org/10.1093/haschl/qxae147
16. Loeffen, A.W.M.; Van Swaaij. B.W.M.; Saminsky, M.; Slot D.E. Common practices of dental implant maintenance among dental hygienists working in the Netherlands - A survey. Int. J. Dent. Hyg.2025, 23(1), 14-25. https://doi.org/10.1111/idh.12841
(point 5) Line 104: the fifth point, (5) “trust building, which aims to establish rapport to improve treatment satisfaction”. May be modified as follows… “gain the patient's trust and approval of the proposed treatment (avoiding understating or overstating the expected results). We are not looking for "treatment satisfaction" but patient satisfaction. This is different and requires adaptation at several levels (socio-economic, mental, physiological, etc.).
(Response to point 5) Thank you for your insightful comments. We have revised the document to reflect your valuable comments:
gain the patient's trust and approval of the proposed treatment
(point 6) Line 119: A note can be added as follows: "Respecting these concepts helps to avoid misunderstandings between the patient and the caregiver which can lead to conflicts that are detrimental to both."
(Response to point 6) We agree with your valuable suggestions and appreciate them. We have added the following to reflect your suggestions:
Respecting these concepts helps to avoid misunderstandings between the patient and the caregiver which can lead to conflicts that are detrimental to both.
(point 7) Line 315: Reference (23) is too old .
(Response to point 7) We agree with the constructive comments. We have provided relevant literature as We referred to “Rogers’ client-centered therapy” as the theoretical basis, but We have added the following relevant literature.
- Lee H,; Chalmers; N.I.; Brow, A. et al. Person-centered care model in dentistry. BMC Oral Health. 2018,18(1), 198-205.
28. Apelian, N.; Vergnes, J.; Hovey, R. et al. How can we provide person-centred dental care?. Br. Dent. J. 2017, 223, 419–424. https://doi.org/10.1038/sj.bdj.2017.806
(point 8) Line 235 : Considering the particularities of the study participants, it seems interesting to specify whether they are general practitioners or specialists in periodontology, implantology, surgery, orthodontics, etc.?
(Response to point 8) Thank you for your insightful comments. Unfortunately, this research study did not survey the research participants’ professional fields of work in the questionnaire. In the discussion section, we presented the limitation that it is necessary to develop a communication competency scale that reflects the job characteristics of dental hygienists. However, since future research is needed to verify the usefulness of PCCS according to the job characteristics of dental hygienists’ workplaces, we added the following as “Limitations and Future Research” in the discussion section:
In addition, it is necessary to confirm the usefulness of the PCCS according to the job characteristics of dental hygienists at each workplace.
(point 9) Lines 301-303: Who are we talking about, to validate the evaluation of the criterion, the correlation between the PCCS and the measurements carried out? Who is concerned with self-efficacy? Who is concerned with job satisfaction? Should the degree of long-term patient satisfaction rather be the indicator of the success of this preliminary investigation?
(Response to point 9) Here is the answer to your concerns about the criterion assessment. The purpose of this study is to self-evaluate the communication skills of dental hygienists and apply them to education and practice. Therefore, I believe that the effectiveness of the PCCS developed through the criterion assessment can be verified based on the theoretical basis that the PCCS is closely related to self-efficacy and job satisfaction.
Reviewer 3 Report
Comments and Suggestions for Authors
Dear authors,
thank you for the opportunity to revise this manuscript.
I recommend the following modifications.
Abstract
Please enrich background and add subparagraphs.
Introduction
“Various communication competency scales have been developed and applied across different healthcare fields": please describe different types of scales, comparing dentistry with the other fields.
M&M
Well conducted methodology, but I suggest to rephrase this paragraph: "Criterion validity was assessed using the 24-item Dental Hygienist Job Satisfaction Scale developed by Park [28]. E...and 0.904 in this study.": maybe it could be useful to provide a scheme/flow-chart to better understand the process of measurement
Results
Please reduce text for EFA description. Please add caption with explanation to Fig.2
Discussion
-"Therefore, providing information at an appropriate level and verifying the patient’s un- derstanding fosters trust and conveys respect [40, 41].": do you think that this can be applied also to other fields? Please add content to discussion.
-"These find- ings align with previous studies that highlight the positive correlation between commu- nication competence, self-efficacy, and job satisfaction [43-46].": please report details of comparison with literature
-please add considerations for future perspectives.
Author Response
Our point-by-point response to the reviewer’s comments and suggestions is listed below:
We thank you for taking the time and effort necessary to review our manuscript and provide us with these valuable comments and suggestions. Accordingly, we revised our manuscript and made changes to it. Please note that changes to the manuscript are highlighted in yellow for your convenience.
Dear authors,
thank you for the opportunity to revise this manuscript.
I recommend the following modifications.
Abstract
(point 1) Please enrich background and add subparagraphs.
(Response to point 1) Thank you for your constructive comments. We have revised and added the background to the abstract as follows:
Communication skills are a core professional competency for dental hygienists. Accordingly, it is vital to develop a systematic scale that can objectively predict the relationship between communication skills and job satisfaction and various psychosocial factors.
Introduction
(point 2) “Various communication competency scales have been developed and applied across different healthcare fields": please describe different types of scales, comparing dentistry with the other fields.
(Response to point 2) We agree with your careful comments and appreciate them. We have added the following in response to your comments.
These include, first, the Global Interpersonal Communication Competence Scale (GICC) [18], which has an adequate number of items to properly reflect the necessary constructs and ensure content validity, second, Nursing Assessment Communication Competence Scale (NACCS) for measuring the communication competency of clinical nurses [19], and third, Communication Ability Scale for assessing the communication skills of medical students and trained simulated patients [20].
M&M
(point 3) Well conducted methodology, but I suggest to rephrase this paragraph: "Criterion validity was assessed using the 24-item Dental Hygienist Job Satisfaction Scale developed by Park [28]. E...and 0.904 in this study.": maybe it could be useful to provide a scheme/flow-chart to better understand the process of measurement
(Response to point 3) Thank you for your insightful comments. We agree with your suggestion and have added a flow chart of the study as follows:
Figure 2
Results
(point 4) Please reduce text for EFA description. Please add caption with explanation to Fig.2
(Response to point 4) We agree with the constructive comments. We have abbreviated the manuscript as follows:
In the first EFA, seven items that exhibited double loadings on multiple factors and two items with factor loadings below 0.4 were removed, resulting in 29 items being retained. In the second EFA, five items that exhibited double loadings on multiple factors and one item with a communality value below 0.4 were removed, resulting in 23 items being retained. In the third EFA, five items that exhibited double loadings on multiple factors and one item with a factor loading below 0.4 were removed, resulting in 17 items being retained. In the fourth EFA, three items that exhibited double loadings on multiple factors and three items with a communality value below 0.4 were removed, resulting in 11 items being retained.
Discussion
(point 5) -"Therefore, providing information at an appropriate level and verifying the patient’s un- derstanding fosters trust and conveys respect [40, 41].": do you think that this can be applied also to other fields? Please add content to discussion.
(Response to point 5) We agree with the meaningful comments and added the need for dental informatics based on prior research in the discussion section as follows:
In particular, in order to strengthen the communication skills of dental hygienists, it is also necessary to operate programs that can increase knowledge of dental information.
(point 6) -"These find- ings align with previous studies that highlight the positive correlation between commu- nication competence, self-efficacy, and job satisfaction [43-46].": please report details of comparison with literature
(Response to point 6) I agree with the meaningful comments and have added details in the discussion section as follows:
Specifically, nurses with high communication self-efficacy showed better performance and job satisfaction in various work tasks [51]. Moreover, communication skills and nursing work environment were identified as important factors influencing nurses' job satisfaction [52], confirming the importance of clinical nurses' communication skills for efficient human resource management [53]. In addition, key components of effective communication skills training in dental education include motivational interviewing, open-ended questioning, affirmations, reflective listening, and summaries to enhance patient engagement and adherence to treatment plans [54].
(point 7) -please add considerations for future perspectives.
(Response to point 7) In response to your constructive comments, I have added the following to the Discussion section:
Time constraints, difficulties in establishing rapport, the oral-health illiteracy of the patients, in addition to poor communication skills, perceptions, and language barriers, often hinder communication between dental hygienists and patients [54].
~~~~~~~~~~~~~~~~~~~
In addition, it is necessary to confirm the usefulness of the PCCS according to the job characteristics of dental hygienists in different workplaces.
Reviewer 4 Report
Comments and Suggestions for Authors
Your manuscript makes a valuable contribution by developing and validating the Patient-Centered Communication Competency Scale (PCCS) for dental hygienists, addressing a critical gap in self-assessment tools for this field. The study’s rigorous methodology and practical relevance are commendable. To further enhance the manuscript for publication, we recommend the following minor revisions:
- Strengthen the introduction with a table or detailed comparison of existing scales and a brief discussion of communication needs across dental hygienist roles.
- Provide more details in the methods section, including the online survey platform, participant recruitment strategies, and Delphi panel expertise.
- Expand the results by analyzing PCCS score differences based on participant characteristics (e.g., experience ≤3 years vs. ≥7 years).
- Enhance the discussion by addressing practical challenges in applying the PCCS (e.g., time constraints) and proposing future research linking PCCS scores with patient feedback.
- Engage a professional editing service to refine repetitive or lengthy sentences, improving academic tone and readability.
- Consider adding a supplementary guide for interpreting PCCS scores or training module examples to boost practical applicability.
These minor revisions will improve the manuscript’s clarity, depth, and impact. We recommend acceptance after minor revisions, as these changes are feasible and will ensure the manuscript meets the journal’s high standards.
The English in the manuscript is generally clear but could benefit from professional editing to enhance conciseness, reduce redundancy, and improve academic tone. Below are specific areas and examples where improvements are needed:
- Repetitive Phrasing:
- Example: Page 1, Line 25: “Communication competency, defined as the ability to interact effectively with others through various communication methods, is considered one of the most essential skills in modern society.”
- Issue: The phrase “communication” is repeated unnecessarily, making the sentence verbose.
- Suggested Revision: “Communication competency, the ability to interact effectively using diverse methods, is a critical skill in modern society.”
- Action: Identify and eliminate redundant terms (e.g., “communication” or “dental hygienists” repeated in close proximity).
- Example: Page 1, Line 25: “Communication competency, defined as the ability to interact effectively with others through various communication methods, is considered one of the most essential skills in modern society.”
- Overly Long Sentences:
- Example: Page 3, Line 103–107: “The key elements of patient-centered communication include: (1) empathic understanding, which involves recognizing and reflecting the patient’s emotions and perspectives; (2) open dialogue, which entails actively listening to patients and providing meaningful feedback; (3) information provision, which focuses on delivering explanations in a clear and accessible manner; (4) shared decision-making, which involves collaborating with patients to determine treatment options; and (5) trust-building, which aims to establish rapport to enhance treatment satisfaction.”
- Issue: The sentence is excessively long and lists multiple complex ideas, reducing readability.
- Suggested Revision: “Patient-centered communication encompasses five key elements: (1) empathic understanding, recognizing and reflecting patients’ emotions; (2) open dialogue, involving active listening and meaningful feedback; (3) clear information provision; (4) shared decision-making with patients; and (5) trust-building to enhance treatment satisfaction.”
- Action: Break long sentences into shorter, clearer ones or use bullet points for lists to improve readability.
- Example: Page 3, Line 103–107: “The key elements of patient-centered communication include: (1) empathic understanding, which involves recognizing and reflecting the patient’s emotions and perspectives; (2) open dialogue, which entails actively listening to patients and providing meaningful feedback; (3) information provision, which focuses on delivering explanations in a clear and accessible manner; (4) shared decision-making, which involves collaborating with patients to determine treatment options; and (5) trust-building, which aims to establish rapport to enhance treatment satisfaction.”
- Lack of Conciseness:
- Example: Page 2, Line 49–51: “Given that communication interventions can have broad and generalized effects across different conditions, evaluating their impact using a consistent analytical framework is essential.”
- Issue: The sentence is wordy and could be more direct.
- Suggested Revision: “Since communication interventions have broad effects, a consistent analytical framework is essential for evaluation.”
- Action: Replace verbose phrases (e.g., “given that,” “can have broad and generalized effects”) with concise alternatives.
- Example: Page 2, Line 49–51: “Given that communication interventions can have broad and generalized effects across different conditions, evaluating their impact using a consistent analytical framework is essential.”
- Inconsistent Academic Tone:
- Example: Page 11, Line 353–354: “However, this study has some limitations. The generalizability of our findings is restricted, as participants were limited to dental hygienists working in clinics and hospitals who communicate directly with patients.”
- Issue: The tone is slightly informal (“some limitations,” “restricted”), and the phrasing could be more precise.
- Suggested Revision: “This study has limitations. Its generalizability is limited, as participants were restricted to dental hygienists in clinics and hospitals who directly communicate with patients.”
- Action: Use precise, formal language (e.g., “limited” instead of “restricted”) and avoid vague qualifiers like “some.”
- Example: Page 11, Line 353–354: “However, this study has some limitations. The generalizability of our findings is restricted, as participants were limited to dental hygienists working in clinics and hospitals who communicate directly with patients.”
- Minor Grammatical Issues:
- Example: Page 7, Line 240: “The corrected item-total correlation coefficient for the developed scale ranged from 0.948 to 0.951.”
- Issue: While grammatically correct, the singular “coefficient” is misleading since it refers to a range across multiple items.
- Suggested Revision: “Corrected item-total correlation coefficients for the scale ranged from 0.948 to 0.951.”
- Action: Ensure subject-verb agreement and clarity in technical descriptions.
- Example: Page 7, Line 240: “The corrected item-total correlation coefficient for the developed scale ranged from 0.948 to 0.951.”
- Overuse of Passive Voice:
- Example: Page 4, Line 139–140: “A preliminary survey was conducted to assess readability, item comprehension, and potential errors in the questionnaire layout.”
- Issue: Passive voice (“was conducted”) reduces dynamism and could be more engaging in active voice.
- Suggested Revision: “We conducted a preliminary survey to assess readability, item comprehension, and questionnaire layout errors.”
- Action: Convert passive constructions to active voice where appropriate to enhance clarity and engagement.
- Example: Page 4, Line 139–140: “A preliminary survey was conducted to assess readability, item comprehension, and potential errors in the questionnaire layout.”
General Recommendation: Engage a professional editing service to polish the manuscript. Focus on reducing redundancy, shortening sentences, and ensuring a consistent academic tone.
Author Response
Our point-by-point response to the reviewer’s comments and suggestions is listed below:
We thank you for taking the time and effort necessary to review our manuscript and provide us with these valuable comments and suggestions. Accordingly, we revised our manuscript and made changes to it. Please note that changes to the manuscript are highlighted in yellow for your convenience.
Comments and Suggestions for Authors
Your manuscript makes a valuable contribution by developing and validating the Patient-Centered Communication Competency Scale (PCCS) for dental hygienists, addressing a critical gap in self-assessment tools for this field. The study’s rigorous methodology and practical relevance are commendable. To further enhance the manuscript for publication, we recommend the following minor revisions:
(point 1) Strengthen the introduction with a table or detailed comparison of existing scales and a brief discussion of communication needs across dental hygienist roles.
(Response to point 1) I agree with the constructive comments. I have added the current status of the role of dental hygienists to the introduction section as follows:
The role of dental hygienists varies depending on the national health care system, legal regulations, and professional recognition of their home country. Although communication is a core competency for dental hygienists, in Korea, it is only explanation-centered due to a lack of job autonomy and an institutional foundation [4]. On the other hand, the United States, Canada, and Northern Europe recognize the communication function of dental hygienists as a core professional competency through a patient-centered and prevention-centered approaches, actively reflecting this paradigm in their education and practice [4].
(point 2) Provide more details in the methods section, including the online survey platform, participant recruitment strategies, and Delphi panel expertise.
(Response to point 2) Thanks for the insightful comments. I've added the data collection method in the methods section as follows:
We randomly selected 50 dental clinics and hospitals in South Korea for inclusion in the study. Through the collaboration of the community dental hygiene society, the survey included an information sheet detailing the study’s purpose, procedure, and estimated completion time.
The content validity of the initial scale items was evaluated using a Delphi survey conducted with an expert panel comprising 10 professors and 8 dental hygienists. The professors are specialists in dental hygiene and dentistry with more than 10 years of work experience, while the dental hygienists hold a master’s degree or higher. All are currently employed in dental clinics and hospitals.
(point 3) Expand the results by analyzing PCCS score differences based on participant characteristics (e.g., experience ≤3 years vs. ≥7 years).
(Response to point 3) I agree with your careful comments. We added the analysis of differences in PCCS by demographic characteristics as a supplementary table and added the results of the analysis by characteristics to the main text. The supplementary table is as follows:
The PCCS level by demographic characteristics was higher for male than for female participants (p = 0.043), while those with a master's degree or higher had a higher level than those with a bachelor's degree or lower (p = 0.001). In addition, PCCS was higher in those aged 40 years or older (p = 0.001).There was no difference in PCCS by length of service, position, or work location, as shown in Table S3.
Table S3. Differences in PCCS by general characteristics
(point 4) Enhance the discussion by addressing practical challenges in applying the PCCS (e.g., time constraints) and proposing future research linking PCCS scores with patient feedback.
(Response to point 4) Thanks for the thoughtful comments. I have added them to the discussion section below:
Time constraints, difficulties in establishing rapport, the oral-health illiteracy of the patients, in addition to poor communication skills, perceptions, and language barriers, often hinder communication between dental hygienists and patients [54].
~~~~~~~~~~
In addition, it is necessary to confirm the usefulness of the PCCS according to the job characteristics of dental hygienists in different workplaces.
(point 5) Engage a professional editing service to refine repetitive or lengthy sentences, improving academic tone and readability.
(Response to point 5) We have received professional editing services to reflect your constructive comments.
(point 6) Consider adding a supplementary guide for interpreting PCCS scores or training module examples to boost practical applicability.
These minor revisions will improve the manuscript’s clarity, depth, and impact. We recommend acceptance after minor revisions, as these changes are feasible and will ensure the manuscript meets the journal’s high standards.
(Response to point 6) I agree with your insightful comments. The PCCS developed in this study is presented in Table S2, but we have additionally provided Korean as follows to enable practical use.
Add Korean to Table S2.
Comments on the Quality of English Language
The English in the manuscript is generally clear but could benefit from professional editing to enhance conciseness, reduce redundancy, and improve academic tone. Below are specific areas and examples where improvements are needed:
Repetitive Phrasing:
Example: Page 1, Line 25: “Communication competency, defined as the ability to interact effectively with others through various communication methods, is considered one of the most essential skills in modern society.”
Issue: The phrase “communication” is repeated unnecessarily, making the sentence verbose.
Suggested Revision: “Communication competency, the ability to interact effectively using diverse methods, is a critical skill in modern society.”
Action: Identify and eliminate redundant terms (e.g., “communication” or “dental hygienists” repeated in close proximity).
Overly Long Sentences:
Example: Page 3, Line 103–107: “The key elements of patient-centered communication include: (1) empathic understanding, which involves recognizing and reflecting the patient’s emotions and perspectives; (2) open dialogue, which entails actively listening to patients and providing meaningful feedback; (3) information provision, which focuses on delivering explanations in a clear and accessible manner; (4) shared decision-making, which involves collaborating with patients to determine treatment options; and (5) trust-building, which aims to establish rapport to enhance treatment satisfaction.”
Issue: The sentence is excessively long and lists multiple complex ideas, reducing
readability.
Suggested Revision: “Patient-centered communication encompasses five key elements: (1) empathic understanding, recognizing and reflecting patients’ emotions; (2) open dialogue, involving active listening and meaningful feedback; (3) clear information provision; (4) shared decision-making with patients; and (5) trust-building to enhance treatment satisfaction.”
Action: Break long sentences into shorter, clearer ones or use bullet points for lists to improve readability.
Lack of Conciseness:
Example: Page 2, Line 49–51: “Given that communication interventions can have broad and generalized effects across different conditions, evaluating their impact using a consistent analytical framework is essential.”
Issue: The sentence is wordy and could be more direct.
Suggested Revision: “Since communication interventions have broad effects, a consistent analytical framework is essential for evaluation.”
Action: Replace verbose phrases (e.g., “given that,” “can have broad and generalized effects”) with concise alternatives.
Inconsistent Academic Tone:
Example: Page 11, Line 353–354: “However, this study has some limitations. The generalizability of our findings is restricted, as participants were limited to dental hygienists working in clinics and hospitals who communicate directly with patients.”
Issue: The tone is slightly informal (“some limitations,” “restricted”), and the phrasing could be more precise.
Suggested Revision: “This study has limitations. Its generalizability is limited, as participants were restricted to dental hygienists in clinics and hospitals who directly communicate with patients.”
Action: Use precise, formal language (e.g., “limited” instead of “restricted”) and avoid vague qualifiers like “some.”
Minor Grammatical Issues:
Example: Page 7, Line 240: “The corrected item-total correlation coefficient for the developed scale ranged from 0.948 to 0.951.”
Issue: While grammatically correct, the singular “coefficient” is misleading since it refers to a range across multiple items.
Suggested Revision: “Corrected item-total correlation coefficients for the scale ranged from 0.948 to 0.951.”
Action: Ensure subject-verb agreement and clarity in technical descriptions.
Overuse of Passive Voice:
Example: Page 4, Line 139–140: “A preliminary survey was conducted to assess readability, item comprehension, and potential errors in the questionnaire layout.”
Issue: Passive voice (“was conducted”) reduces dynamism and could be more engaging in active voice.
Suggested Revision: “We conducted a preliminary survey to assess readability, item comprehension, and questionnaire layout errors.”
Action: Convert passive constructions to active voice where appropriate to enhance clarity and engagement.
General Recommendation: Engage a professional editing service to polish the manuscript. Focus on reducing redundancy, shortening sentences, and ensuring a consistent academic tone.
(Response on the Quality of English Language) We sincerely appreciate your thoughtful English proofreading comments. We received English proofreading from Editage,
Round 2
Reviewer 2 Report
Comments and Suggestions for Authors The authors have updated the bibliography and addressed the questions raised.However, it is a shame to maintain abbreviations in the abstract,
which hinders the understanding of the abstract for readers who will not read the article.